

# A critical issue in model-based inference for studying trait-based community assembly and a solution

Cajo J.F. ter Braak[1], Pedro Peres-Neto[2] and Stéphane Dray[3]

[1] Biometris, Wageningen University & Research, Wageningen, The Netherlands
[2] Department of Biology, Concordia University, Montreal, Canada
[3] Laboratoire de Biométrie et Biologie Evolutive, Université Claude Bernard (Lyon I), Villeurbanne, France

Corresponding author
Cajo J.F. ter Braak,
cajo.terbraak@wur.nl

## ABSTRACT

Statistical testing of trait-environment association from data is a challenge as there is no common unit of observation: the trait is observed on species, the environment on sites and the mediating abundance on species-site combinations. A number of correlation-based methods, such as the community weighted trait means method (CWM), the fourth-corner correlation method and the multivariate method RLQ, have been proposed to estimate such trait-environment associations. In these methods, valid statistical testing proceeds by performing two separate resampling tests, one site-based and the other species-based and by assessing significance by the largest of the two $p$-values (the $p_{max}$ test). Recently, regression-based methods using generalized linear models (GLM) have been proposed as a promising alternative with statistical inference via site-based resampling. We investigated the performance of this new approach along with approaches that mimicked the $p_{max}$ test using GLM instead of fourth-corner. By simulation using models with additional random variation in the species response to the environment, the site-based resampling tests using GLM are shown to have severely inflated type I error, of up to 90%, when the nominal level is set as 5%. In addition, predictive modelling of such data using site-based cross-validation very often identified trait-environment interactions that had no predictive value. The problem that we identify is not an "omitted variable bias" problem as it occurs even when the additional random variation is independent of the observed trait and environment data. Instead, it is a problem of ignoring a random effect. In the same simulations, the GLM-based $p_{max}$ test controlled the type I error in all models proposed so far in this context, but still gave slightly inflated error in more complex models that included both missing (but important) traits and missing (but important) environmental variables. For screening the importance of single trait-environment combinations, the fourth-corner test is shown to give almost the same results as the GLM-based tests in far less computing time.

## INTRODUCTION

According to the habitat templet theory (*Southwood, 1977*; *Townsend & Hildrew, 1994*), evolution selects for species characteristics (i.e., traits) appropriate to their environment. Such traits influence community assembly (*Ackerly & Cornwell, 2007*) and a major goal in contemporary ecology has become to identify among a set of traits which ones interact with the environment and which do not. Although most traits may influence the way species are distributed in space, not all environmental features necessarily select for these traits, hence the need to test for trait-environmental interactions and the need to select only the relevant trait-environment combinations in predictive models.

The first statistical methods to uncover and describe trait-environment associations were correlation-based with as prime examples the Community Weighted trait Means (CWM) approach (*Lavorel et al., 2008*) that correlates community weighted means with environmental features, the fourth-corner correlation (*Dray & Legendre, 2008*; *Legendre, Galzin & Harmelin-Vivien, 1997*) and the multivariate method RLQ (*Dolédec et al., 1996*; *Dray et al., 2014*). See *Kleyer et al. (2012)* for a review of methods. Notwithstanding the availability and wide use of these methods, it took some time to understand the behaviour of these methods and to develop valid statistical tests to assess trait-environment associations. *Dray & Legendre (2008)* showed that randomization tests based on either site or species permutations lead to increased type I error rates. The issue of increased type I error rate was solved by *Ter Braak, Cormont & Dray (2012)* and *Peres-Neto, Dray & ter Braak (2016)* who showed that regardless of the method used to assess trait-environment relationships, valid statistical testing requires both a site-based and a species-based analysis, each resulting in a $p$-value. They showed that correct rates are achieved by assessing significance by the largest of the two $p$-values (the $p_{max}$ permutation test). If it is desired to account for phylogenetic relationships among the species and/or spatial and temporal correlations among the sites, the random permutations should be replaced by restricted permutation or bootstrap (*Lapointe & Garland, 2001*; *Wagner & Dray, 2015*), but the principle of the $p_{max}$ test remains unchanged.

More recently, regression-based methods have been proposed for studying trait-environment relations (*Brown et al., 2014*; *Cormont et al., 2011*; *Jamil et al., 2013*; *Pollock, Morris & Vesk, 2012*; *Warton et al., 2015*; *Warton, Shipley & Hastie, 2015*). These methods model the abundance (or presence–absence) of multiple species across sites (communities) as a function (linear or non-linear) of species traits and environmental variables. If these are generalized linear (mixed) models, main effects of traits and environmental variables and their interactions are specified on a link-scale, for count data usually the log-scale giving a log-linear model. Interaction terms represent trait-environment associations, each being a product of a single trait with a single environmental variable. The associated (standardized) regression coefficients provide insights regarding the strength and direction of trait-environment associations (positive and negative associations, e.g., large-bodied species tend to occur more often in low-temperature environments than in high-temperature ones, known as Bergmann's rule (*Bergmann, 1847*).

The main effects in the GLM model represent the separate effects of environmental variables and traits on species distributions. For example, traits might be used to predict the distribution of species in an 'average' environment; by adding the environmental effects and their interaction with the traits, the distribution of a given species in a specific environment may be predicted. Also, by setting main effects to be polynomial terms of quantitative trait and environment variables (or, simply, to factors for species and site) the model includes the simplest model for ecological niches, which shows Gaussian species response to the environmental variable and has equal niche breadths (*Jamil et al., 2013*; *Jamil & ter Braak, 2013*; *Ter Braak & Looman, 1986*). If the environmental optima of species in this model are related to their traits, the trait-environment relationships are exactly represented by the interaction terms, all being a product of a given environmental variable and a given trait. In regression-based analyses of trait-environment relations, neither the trait nor the environment takes the role of response variables; instead they are predictor variables in a model in which the response variable is the abundance (or presence–absence) of species in sites.

In this paper, we focus on the performance of statistical tests to assess the significance of interaction terms in GLM-based methods as these represent trait-environment relations. Different testing procedures have been used so far. *Cormont et al. (2011)* fit a linear trait-environment model (LTE) with interactions to log-transformed species data and used the $p_{\max}$ permutation test of *Ter Braak, Cormont & Dray (2012)* for statistical testing of the interaction. *Jamil et al. (2012)* implemented a mixed-model approach with fixed trait-environment coefficients and random species-by-environment coefficients; interaction terms were selected via a tiered forward-selection approach and their significance was assessed by likelihood ratio tests. *Pollock, Morris & Vesk (2012)* and *Jamil et al. (2013)* moved away from 'least-squares after data transformation' by using generalized linear mixed models (GLMM) for count or presence–absence data. *Brown et al. (2014)* and *Warton, Shipley & Hastie (2015)* developed the same model under a simpler GLM framework without the extra random terms for species, and complemented it with resampling techniques, namely cross-validation for selection and bootstrapping of sites for statistical testing. Their method is thus essentially a community-level analysis in that the site (community) is the statistical unit on which the statistical inference is based. Their method of statistical testing is implemented via the functions 'traitglm' and 'anova.traitglm' in the R package 'mvabund' (*Wang et al., 2012*). These GLM-based methods are quite promising because they use mainstream statistical methods, allowing model selection, prediction and standard diagnostics for goodness-of-fit as well as greater flexibility in accounting for additional structure in the data, compared to previous correlation-based approaches.

Conceptually, the most important difference between these GLM and GLMM approaches is that *Brown et al. (2014)* and *Warton, Shipley & Hastie (2015)* solve the problem of residual correlation among species in their GLM by resampling sites and ignore any species-level random effects. Conversely, *Pollock, Morris & Vesk (2012)* and *Jamil et al. (2013)* account for randomness at the species-level in their GLMM, but have either no, or only a very simplistic, method to account for residual correlation among species, respectively. In practice, GLM is much quicker and numerically more stable than GLMM, so that using

GLMM with bootstrapping is not yet very appealing, particularly when many environmental variables need to be analyzed simultaneously.

Whereas the performance of statistical tests associated with the fourth-corner correlation has been well evaluated in the literature (*Dray & Legendre, 2008*; *Peres-Neto, Dray & ter Braak, 2016*; *Ter Braak, Cormont & Dray, 2012*), no study has been designed to evaluate and compare the type I error rate and power of statistical tests on trait-environment interaction in the GLM framework. One obvious question is how the simpler GLM model performs when the data actually follow a log-linear model with random effects, i.e., a GLMM model (*Jamil et al., 2013*). Is it then sufficient to simply resample (bootstrap or permute) sites or residuals associated with sites, which is similar to a community-level analysis, or is there also a need for a species-level analysis (resampling species or residuals associated with species) as found for correlation-based approaches, and, if so, how can we then explain the difference in outcome between the community-level and species-level analyses?

In this paper we investigate these questions by simulating data according to models with and without trait-environment interaction and re-analysing a literature data set as an illustrative case study. We apply four statistical tests which differ in the way resampling is performed (resampling sites, species or both) and how the test statistic is calculated (assuming a negative-binomial or a Poisson distribution). We report on the observed type I error rates of these procedures in the data sets simulated without trait-environment interactions and their power in simulated data sets with this interaction. We also apply the predictive modelling approach of *Brown et al. (2014)* to see how many times trait-environment terms were falsely judged predictive when there was in fact no interaction between the observed traits and the environment. The simulated data were based on Gaussian response models (*Dray & Legendre, 2008*; *Peres-Neto, Dray & ter Braak, 2016*) and log-linear models with random effects (i.e., GLMM models).

## THEORY AND METHODS

### Statistical models for trait-environment interaction

The data required for the statistical analyses in this paper are abundance data of $m$ species across $n$ sites, together with trait data on the $m$ species, and environmental data on the $n$ sites. In its simplest form, there is one quantitative trait $\mathbf{t}$ (with $m$ values $\{t_j\}$, $j = 1, \ldots, m$) and one quantitative environmental variable $\mathbf{e}$ (with $n$ values $\{e_i\}$, $i = 1, \ldots, n$). The abundance of species $j$ in site $i$, denoted by $y_{ij}$, is assumed to be (similar to) a count that follows a negative-binomial distribution with mean $\mu_{ij}$ and unknown overdispersion. The statistical model that we use for detecting the trait-environment interaction is the GLM model

$$\log\left(\mu_{ij}\right) = R_i + C_j + b_{te} t_j e_i, \tag{1}$$

where $b_{te}$ is the coefficient measuring direction and strength of the trait-environment (**t**-**e**) interaction and $R_i$ and $C_j$ are the main effects for site $i$ and species $j$, respectively. The main effects are thus formed by factors for site and species and can thus approximate any non-linear function of $\mathbf{e}$ and $\mathbf{t}$, respectively. In total, the model has $n + m + 2$ unknown parameters (including the unknown overdispersion).

An alternative for the GLM model is the GLMM model, which we present here for later reference,

$$\log(\mu_{ij}) = R_i + C_j + \beta_j e_i \text{ with } \beta_j = b_0 + b_{te} t_j + \varepsilon_{\beta j} \tag{2}$$

with $\beta_j$ a species-specific slope with respect to $\mathbf{e}$, modelled as a linear model of trait $\mathbf{t}$, with intercept $b_0$ and slope $b_{te}$, and with $\varepsilon_{\beta j}$ a normally distributed error term with mean 0 and variance $\sigma_\beta^2$. By inserting the model for $\beta_j$ in the log-linear model, we see that $b_{te}$ is indeed the coefficient of the interaction $t_j e_i$. The term $b_0 e_i$ can be absorbed in the row main effect $R_i$, and $\varepsilon_{\beta j} e_i$ represents additional species-specific random variation that interacts with the observed environment.

## Statistical tests on trait-environment interaction

The trait-environment interaction in Eq. (1) can be tested by fitting the model with and without the interaction term, the latter by setting $b_{te} = 0$, calculating the likelihood ratio (LR) statistic of the two models for the data and assessing its significance via resampling (*Warton, Shipley & Hastie, 2015*). We consider four ways of carrying out the test. The first test uses the LR based on a negative-binomial response distribution and is therefore rather slow. To investigate whether we could improve on speed without sacrificing the type I error rate and loosing (too much) power, we set the response distribution to Poisson in the other three tests, as any issue due to overdispersion is accounted for by the resampling procedure. Moreover, theory tells that Poisson likelihood is the only likelihood for non-negative data and models that gives consistent estimates under misspecification of the distribution; the normal likelihood/least-squares has this feature for unbounded data (*Gourieroux, Monfort & Trognon, 1984a*; *Gourieroux, Monfort & Trognon, 1984b*; *Wooldridge, 1999*). In detail, the four tests are:

1. **anova.traitglm.** The first test uses the 'anova.traitglm' function of version 3.11.5 of the R package mvabund (*Wang et al., 2012*) with site-based resampling and calculation of the LR assuming a negative-binomial distribution (as in the data-generating models of 'Statistical models for trait-environment interaction' and 'Simulation models'). This function resamples sites by bootstrapping probability integral transform (PIT) residuals with all residuals across species of the same site kept together. The code for obtaining the significance of the interaction is simply:

   ```
   model1 <- traitglm(Y,E,T, composition=TRUE)
   anova.traitglm(model1, nBoot=nBoot)
   ```

   where $\mathbf{Y}$, $\mathbf{E}$ and $\mathbf{T}$ contain the abundance data $y_{ij}$, the environmental values $\mathbf{e} = e_i$ and the trait values $\mathbf{t} = t_j$, respectively. In the many-traits case, $\mathbf{T}$ contains the $q$ simulated traits. The number of bootstrap samples is set to 39. With the observed sample this gives 40 samples and a minimum $p$-value of $1/40 = 0.025$. This number is sufficient at the nominal level of 5% of the test that we used, as Monte Carlo significance tests are unbiased for any number of re-samples (*Hope, 1968*). We could increase the number of bootstraps to obtain a small increase in power.

2. **sites.** The second test also resamples sites, but differs from the first test in that it permutes sites (instead of bootstrapping them) and calculates the Poisson LR (instead of the negative-binomial LR). For statistical testing (rather than estimation), we prefer permutation to bootstrapping. Therefore, we wrote special purpose functions in R using glm on vectorised abundance data **Y** for calculating the LR statistic assuming Poisson distributed abundance data. When permuting sites, the values in **e** are permuted instead of the rows of the abundance table **Y** (Appendix S2). We used 39 permutations to make the test most similar to the anova.traitglm test.

3. **species.** This test is similar to the previous one but permutes species (instead of sites) and calculates the Poisson LR.

4. **max r/c.** The fourth test applies both tests 2 and 3 and takes the maximum of their $p$-values. It thus a GLM-based $p_{\max}$ test. The 'r/c' is a mnemonic for rows/columns (corresponding to sites/species).

## Simulation models

Abundance data on $m = 30$ species in $n = 30$ sites were generated by two simulation models, a log-linear simulation model and a one-dimensional Gaussian response model, detailed in Appendix S1. In summary, two traits **t** and **z** (both $m$ values) and two environmental variables **e** and **x** (both $n$ values) were drawn independently from the standard normal distribution; **t** and **e** are taken as the observed trait and the observed environment, respectively, and **z** and **x** as unobserved. As such, variables **z** and **x** are latent variables that are unrelated to (i.e., independent of) the observed ones; alternatively, **z** and **x** represent simply noise, more specifically, variability among species in their response to the environment, as in GLMM models (*Jamil et al., 2013*; *Pollock, Morris & Vesk, 2012*), and unobserved variability among sites, respectively. Either way, unobserved variation is likely the case in most ecological data and needs to be considered more often in simulation studies.

The statistical test procedure seeks to detect whether the observed trait **t** and the observed environment **e** are associated (i.e., interact), without knowledge about the two latent variables **z** and **x**, as these are unobserved/unmeasured. In the null models, abundance data are generated without any interaction between the observed **t** and **e**, but with an interaction between, for example, the unobserved trait **z** and the observed environmental variable **e**.

In the Gaussian response model, this is achieved by generating the expected abundance $\mu_{ij}$ of species $j$ at site $i$ as a Gaussian response function of **e** and **z**:

$$\mu_{ij} = h_j \exp\left[-\frac{(e_i - z_j)^2}{2\sigma_j^2}\right], \tag{3}$$

where $h_j$ is the maximum value and $\sigma_j$ is the tolerance or niche breadth of species $j$ that are both constants or random, with $\sigma_j$ independent of **t**. As **z** and $\sigma_j$ are independent of the observed trait **t**, this model by definition contains no association between **t** and **e**. This is the 'environmentally structured but trait random' case, in short the 'trait random' case. Similarly, we can define a 'trait structured but environment random' case, in short the

'environment random' case, by replacing $e_i$ by $x_i$ and $z_j$ by $t_j$, and a 'both random' case by replacing $e_i$ by $x_i$ in Eq. (3).

The corresponding log-linear simulation model has free main effects ($R_i$ and $C_j$) and one interaction, namely either $z_j e_i$ or $t_j x_i$ respectively, e.g., for the 'trait random' case

$$\log(\mu_{ij}) = R_i + C_j + b_{ze} z_j e_i \qquad (4)$$

with $b_{ze}$ the coefficient measuring the direction and strength of the **z**-**e** interaction, and similarly for other interactions later on. The interaction $t_j e_i$ is missing, so there is no association between **t** and **e** in the model. Equation (4) is like Eq. (3), the model of a GLMM with random species-specific slopes with respect to the environment **e** (*Jamil et al., 2013*), but without **t**–**e** interaction ($b_{te} = 0$).

The Gaussian model of Eq. (3) and log-linear model of Eq. (4) are closely related, at least when the tolerance is constant ($\sigma_j = \sigma$). First, the maximum $h_j$ does not depend on the trait and thus $\log(h_j)$ can be absorbed in the free coefficient $C_j$ in Eq. (4). Working out the square in Eq. (3) gives squared terms in $e_i$ and $z_j$ that can be absorbed in the free main effects of Eq. (4) and a product $z_j e_i$ with associated coefficient $1/\sigma^2$, which is then equal to $b_{ze}$ in Eq. (4). With variable niche breadths, the Gaussian response model is not an easy GLM.

The log-linear model allows one case that is not available in the one-dimensional Gaussian model. This case has two interaction terms: **t** interacts with **x** and **e** interacts with **z**, but **t** does not interact with **e**,

$$\log(\mu_{ij}) = R_i + C_j + b_{ze} z_j e_i + b_{tx} t_j x_i. \qquad (5)$$

These are simple models. In the reported simulations, we also included structured and unstructured noise in the expected abundances, respectively (Appendix S1); these are important to make the data more realistic, but are not essential for our main results.

Abundance data $y_{ij}$ ($i = 1, \ldots, n; j = 1, \ldots, m$) were drawn from the negative-binomial distribution with mean $\mu_{ij}$ and overdispersion parameter 1, giving variance function $\mu_{ij} + \mu_{ij}^2$.

To mimic a situation with many traits and a single environmental variable, $t_j$ in the above equations was taken as the sum of $q$ independent traits, normalized to a theoretical variance of 1.

To study the power of different methods to assess parameter significance, once the type I error had been controlled, the interaction between the observed trait and environment, i.e., the term $b_{te} t_j e_i$, was added to the log-linear model with non-zero regression coefficient $b_{te}$, and $z_j$ was replaced by $t_j$ in the Gaussian response model. Each model was simulated 1,000 times.

## Predictive modelling by lasso

*Brown et al. (2014)* proposed selecting trait-environment interactions on the basis of their predictive power using GLM with the least absolute shrinkage and selection operator (lasso). This approach uses penalized regression and a key point is the selection of the

penalty parameter; if it is set very high, no predictor variable enters the model, if it is set to 0 all variables enter. The penalty is usually selected by cross-validation. *Brown et al. (2014)* used a site-based cross-validation approach, treating species as fixed factors. To evaluate this approach, we simulated 1000 data sets as in the 'trait random' case of the Gaussian model with $n = m = 30$ and 10 traits and a single environmental variable. We applied the function traitglm of the R package mvabund with arguments method = "cv.glm1path" and composition = TRUE, then counted the number of simulated data sets in which the best model, i.e., that gave the "best predictive performance" (*Wang et al., 2012*), contained any non-zero trait-environment coefficients. As a control experiment, we analyzed data from a model without any interaction of a latent trait with the observed variables. We also applied species-based cross-validation to the same data sets to determine whether this led to similar results.

## Case study

As an illustration, we re-analysed the data of *Choler (2005)* who investigated the shift in Alpine plant traits along a snow-melt gradient. This is the aravo data set in the R package ade4 (*Dray & Dufour, 2007*). It has 82 species and 75 sites. Here we considered only one trait and one environmental variable, namely Spread (maximum lateral spread of clonal plants) and Snow (mean snowmelt date in Julian day averaged over 1997–1999). Because the distribution of Spread is right-skewed, it was logarithmically transformed before analysis.

First, we performed statistical tests based on the squared fourth-corner correlation, using the R package ade4 (*Dray & Dufour, 2007*). Second, we performed regression-based analyses as described in *Brown et al. (2014)* and *Warton, Shipley & Hastie (2015)*. In such analyses, the abundance of the species is the response variable for which we must specify a distribution. We used the default distribution of the R package mvabund (*Wang et al., 2012*), the negative-binomial distribution, and checked whether this distribution is tenable for these data. Following *Warton, Shipley & Hastie (2015)*, we did this by plotting the Dunn-Smith residuals of the model 'site+species+species:Snow' against the fitted values. The plot (Fig. S1) does not show any clear pattern that would suggest an ill-specified mean–variance relation. We therefore continued analysis with this default distribution. We then applied the anova.traitglm test and the other three tests in 'Statistical tests on trait-environment interaction.' We used 999 resamples in each test. Third, as an alternative to statistical testing, we applied predictive modelling as in 'Predictive modelling by lasso.' With a single trait-environment interaction, the choice is between a model with and a model without the interaction.

## RESULTS

### Statistical testing results

In the Gaussian model simulations, the 'trait random' case is the only one that generates a significantly inflated type I error rate (>0.40) in the anova.traitglm test, whereas the max r/c test provides appropriate rates (Table 1). With six species and five or ten random traits, the type I error rate of anova.traitglm was even more inflated (>0.75), and for twenty species with 10–30 traits the type I error rate was even greater than 0.90.

Table 1  Type I error rates of the anova.traitglm test (site-based bootstrap approach based on the negative-binomial likelihood) and the max r/c test ($p_{max}$ test based on the Poisson likelihood) for 1,000 simulated negative-binomial data sets using the Gaussian response model and $n = m = 30$. Value in bold represents the inflated error compared to the nominal level of 0.05.

| Test procedure | Scenario | | |
| --- | --- | --- | --- |
| | Trait random (z and e) | Environment random (t and x) | Both random (z and x) |
| anova.traitglm | **0.417** | 0.055 | 0.045 |
| max r/c | 0.047 | 0.055 | 0.019 |

In the log-linear simulation model, similar inflation of type I error rates is found. The type I error rate of the anova.traitglm test increases with the size of the coefficient $b_{ze}$ in Eqs. (4) and (5) (Fig. 1, grey lines) up to values around 0.50. The sites test results in similar $p$-values. The species test results in much lower $p$-values than the sites test when $b_{tx} = 0$, but similar $p$-values if $b_{tx} = b_{ze}$ (solid and dotted blue lines in Fig. 1, respectively). The latter is expected as, when $b_{tx} = b_{ze}$, the model is perfectly symmetric in species and sites. The max r/c test, which takes the maximum of the $p$-values per simulated dataset, results in $p$-values around or below 0.05 if $b_{tx} = 0$ but has a slightly inflated type I error rate if $b_{tx} = b_{ze} > 0.4$ (solid and dotted black lines in Fig. 1, respectively). The highest type I error rate of the $p_{max}$ test is 0.08 (Fig. 1).

Figure 2 shows the power of max r/c test. Recall that power here is estimated at a higher than 0.05 nominal level because the test does not fully control the type I error. The actual size of the test is 0.08 (the maximum observed type I error rate for this test in Fig. 1). As expected, power increases with the trait-environment coefficient $b_{te}$, but decreases with increasing $b_{ze}$, which represents either the interaction of **e** with an unmeasured trait or noise in the model for the species-specific slopes in Eq. (2). The effect of an additional noise component (modelled by $b_{tx}$) on power is not very large, except near $b_{te} = 0$, where power is in fact type I error rate, which was inflated when $b_{ze} > 0.4$, as in Fig. 1.

## Predictive modelling results

In the 'trait random' case, site-based cross-validation to select the penalty parameter of the lasso often resulted in models with trait-environment terms that have, in fact, no predictive value; the number of data sets with such terms was 851 (in 1,000 simulated data sets) with 463 data sets giving one interaction coefficient that was at least 0.1 in absolute value. In the control experiment using a model without any interaction of a latent trait with the observed variables, we found low numbers of false positive (<20 out of 1,000). With $m = 10$ (instead of 30), there were 855 false positives numbers with 710 and 403 data sets with a coefficient larger than 0.1 and 0.3, respectively. Applying species-based cross-validation resulted for $m = 30$ in two data sets with nonzero trait-environment coefficients and for $m = 10$ in ten such data sets.

## Case study results

Statistical tests on the interaction between Spread and Snow using the fourth-corner correlation resulted in $p$-values of 0.002 and 0.361 for the site- and species-based

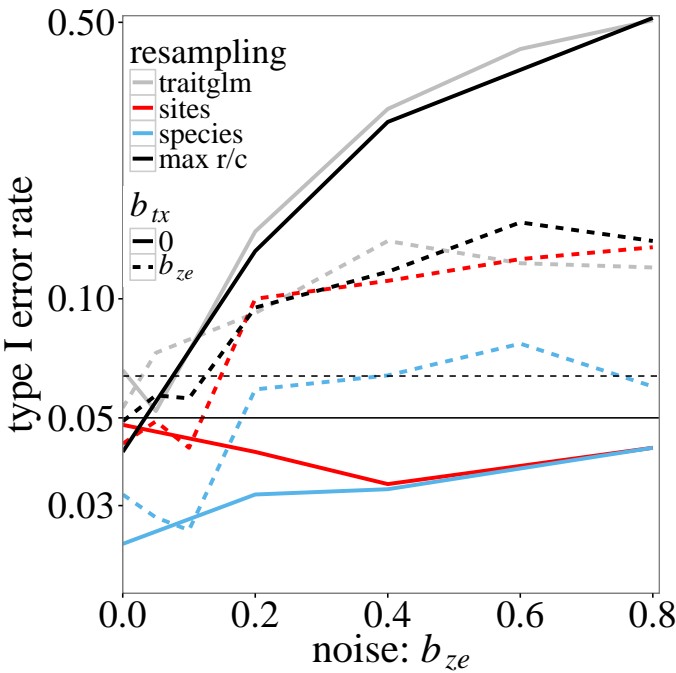

**Figure 1** **Type I error rate of the four tests of 'Statistical tests on trait-environment interaction' on the trait-environment (t–e) interaction in the log-linear simulation model of Eq. (5) in relation to the size of the z–e nuisance interaction (of e with a latent trait z): anova.traitglm (site-based bootstrap with negative-binomial likelihood) and sites (site-based permutation with Poisson likelihood), species (species-based permutation with Poisson likelihood), and max r/c (GLM based $p_{max}$ test that combines the sites and species tests).** The t–x nuisance interaction (of t with a nuisance environmental variable x) is either absent (solid lines) or equal to the size of the z–e nuisance interaction (dashed lines). The vertical scale is logarithmic. The data were generated using a negative-binomial distribution with variance function $\mu_{ij} + \mu_{ij}^2$. The horizontal line at 0.05 indicates the nominal significance threshold; error rates above the dotted line (at 0.064) are significantly greater than 0.05.

permutation tests, respectively (999 permutations). In the $p_{max}$ test, these $p$-values are combined by taking their maximum and the final $p$-value was thus 0.361. The conclusion from the fourth-corner analysis is that there is no evidence that Spread and Snow are associated.

Using GLM-models, the site-based approaches found evidence for interaction between Spread and Snow ($p$-value of 0.001 for both the anova.traitglm and sites tests). By contrast, the species and max r/c tests did not detect evidence for this interaction ($p$-value of 0.376 for both).

Site-based cross-validation in lasso-based predictive modelling selected the model with interaction, with a fourth-corner regression coefficient of 0.066 with respect to the standardized trait and the standardized environment variable. Species-based cross-validation selected the model without interaction.

## DISCUSSION

Our simulations showed that resampling of sites is simply not sufficient to generate a valid statistical test on trait-environment association under a GLM framework. This holds true

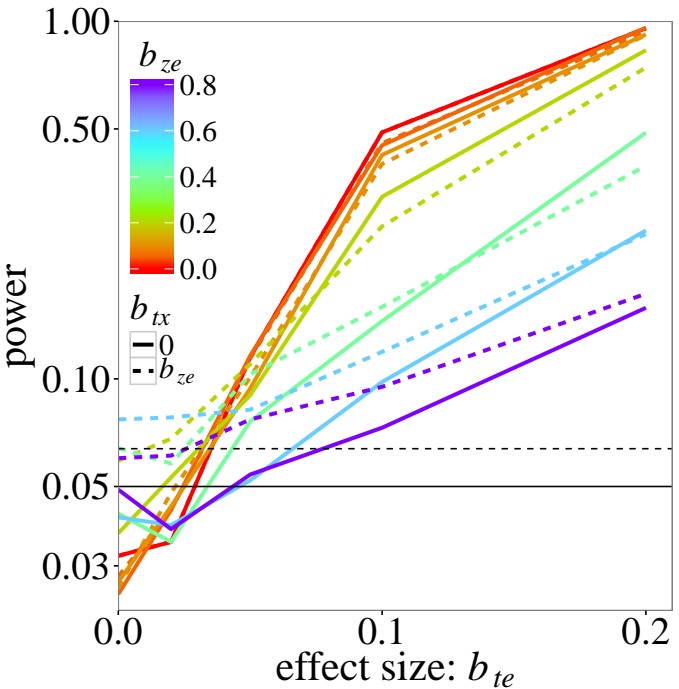

**Figure 2** Power of the max r/c test (GLM based $p_{max}$ test) on the trait-environment (t–e) interaction, added to the log-linear simulation model of **Eq. (5)**, in relation the size of the interaction of interest with colours for various sizes of the z–e nuisance interaction (of e with a latent trait z) and linetypes as in **Fig. 1**. The vertical scale is logarithmic. The data were generated using a negative-binomial distribution with variance function $\mu_{ij} + \mu_{ij}^2$, whereas the test statistic was based on the Poisson likelihood. The horizontal line at 0.05 indicates the nominal significance threshold; rates above the dotted line (at 0.064) are significantly greater than 0.05.

not only for advanced bootstrap methods, such as is implemented in the anova.traitglm function of the R package mvabund (*Wang et al., 2012*), but also for permutation tests using either GLM (Fig. 1) or simple methods such as CWM and fourth-corner (*Peres-Neto, Dray & ter Braak, 2016*). The reason is that sites are not the only random statistical units; species are random units as well (*Ives & Helmus, 2011*), particularly when we want to determine which traits interact with particular environmental features in conditioning species distributions. Species and sites form two crossed random factors and both need to be considered in models. A GLM without resampling would account only for the residual variation (technically, the *nm* independent response distributions of the *nm* abundance values). A GLMM model is able to account for more levels of randomness. A GLMM model for trait-environment association accounts for the randomness of species responses to the environment and for how traits modulate these responses. These levels of randomness are not considered in GLMs; therefore, these will not provide valid statistical inference (see Appendix S3 for an explicit explanation). An escape is to mimic the randomness and discover the relevant variation by resampling. Therefore, we propose in this paper to resample not only sites but also species and to combine the two tests in the $p_{max}$ procedure. Note that a test using simultaneous resampling of species and sites, as originally proposed for RLQ (*Dolédec et al., 1996*), does not control the type I error (*Dray & Legendre, 2008*);

instead resampling needs to be performed independently for sites, which fixing species, and vice-versa for species.

All site-based analyses of the case study data found evidence for Spread-Snow interaction, but the species-based analysis, and thus the $p_{max}$ test, did not. Why do site-based and species-based analyses differ with these data? An explanation is that there is an 'unobserved' trait, namely Specific Leaf Area (SLA), that is almost uncorrelated with Spread (correlation 0.02) and that strongly interacts with Snow as both the site-based and species-based tests are highly significant using either the fourth-corner correlation or the GLM tests using Poisson log-likelihood (all $p < 0.001$, yielding $p_{max} < 0.001$). From this, we thus conclude that there is evidence that SLA interacts with Snow and no evidence that Spread interacts with Snow. The site-based resampling tests are not able to detect the large variation among the species-specific regression coefficients in Eq. (2). In the GLM fit, Spread explains an amount of variation that is easily explainable by chance. The conclusion is that the GLM-based tests require not only site-based resampling but also species-based resampling.

One might think that the problem with site-based resampling is just an 'omitted variable bias' problem (*Robins, Rotnitzky & Zhao, 1994*), also known as a 'confounding variable' problem (*McDonald, 2014*), which leads to biased regression coefficients and thus biased inference. This problem identifies a well-known fundamental limitation of all regression models and occurs when omitting a variable that is both correlated to the response and to the predictor. Our case is different in that the omitted variable is uncorrelated to the variables under investigation, the trait and environmental variables. In addition, the problem we identify is not bias in the regression coefficient but a wrong (too low) standard error of estimate. Instead of viewing it as a case of "omitted variable bias", we consider it as a case of "ignoring random effects," which here means ignoring species as a random factor.

The $p_{max}$ procedure was shown via simulations to control the type I error in simple models for trait-environment association, such as the ones used in the context of fourth-corner correlation (*Dray et al., 2014*; *Peres-Neto, Dray & ter Braak, 2016*) and GLMM (*Jamil et al., 2013*; *Pollock, Morris & Vesk, 2012*). However, the present study assessed these models also under more complex (possibly more realistic) scenarios including multiple latent factors. We show that if both the trait and the environmental variables interact with an important latent environmental variable and an important latent trait, respectively, the control over type I error is faulted. That said, in all simulations we carried out (with up to 100 species and sites), we never observe a type I error rate above 0.10. In even more complex scenarios, however, the inflation could possibly be more severe. At this point, as long as nothing better is proposed, our advice is to divide the nominal level in the $p_{max}$ test by two (e.g., $0.05/2 = 0.025$) for species and site numbers between 20 and 100, so as to control empirically the type I error at a level of 0.05. This is akin to what happens in Bonferroni-correction for multiple testing (*Verhoeven, Simonsen & McIntyre, 2005*). With numbers of both species and sites below 20, based on our results, correction is not necessary. A more precise correction factor for any specific number of species and sites could be determined via simulation of data using models in which the interactions between the traits and the environment are set to zero.

The fact that the $p_{\max}$ test does not fully control the type I error in Eq. (5) should not come as a surprise. The test considers only whether there is a link between the species distributions and the environment$(\mathbf{Y}\leftrightarrow\mathbf{e})$ and a link between the species distributions and the trait $(\mathbf{Y}\leftrightarrow\mathbf{t})$ at the nominal significance level. What these links entail is not part of the theory behind the test (*Ter Braak, Cormont & Dray, 2012*). For example, by taking a test statistic that is sensitive to main effects, one could easily establish these links. For detecting trait-environment association, the test statistic should not be sensitive to main effects and be able to detect interactions only. There is a more fundamental issue, however. In the GLM, trait-environment association is an interaction, but interaction is link-dependent. An extreme example of this is that main effects in a log-linear model are products (i.e., interactions) in a linear model (with identity link). With binary data, a GLM is often used with a logistic link function, generating implicitly another definition of interaction. Seldom is a logarithmic link used as it leads to numerical issues. The complementary log–log link function is the one that is closest to the count scale and is perhaps the most appropriate for trait-environment modelling. In Eq. (5), trait $\mathbf{t}$ and the environmental variable $\mathbf{e}$ are both important in the species-site interaction space of the log-linear model. One could argue that this model shows some form of $\mathbf{t}$–$\mathbf{e}$ association, namely when analyzed with any other link function than the logarithmic one. The same would happen the other way round, i.e., when the logarithm in the generating model is replaced by a slightly different function (reality being not precisely log-linear), but the analysis uses the log-link. Hence, model misspecification could lead to different (and possibly wrong) interpretations of trait-environment interactions.

We made the computation quicker by changing the negative-binomial likelihood to the Poisson likelihood. This did not appear to affect the power of the test much, as observed in the comparison of the anova.glm and sites tests in Fig. 1. We used random permutations in our tests. In future applications, this may need further adaptation as sites may be structured in space (spatial autocorrelation) and time (temporal autocorrelation) and species form a phylogeny (phylogenetic autocorrelation), so that neither sites nor species are really completely independent or exchangeable units. The net effect will be that the effective number of units (i.e., degrees of freedom) is actually smaller than the number observed in the data, likely generating a liberal test when random permutations are used. Possible alternatives for random permutations are restricted permutations (*Lapointe & Garland, 2001*) or data simulation that keeps the original spatial or phylogenetic structure in data (*Wagner & Dray, 2015*). Note also that the need of resampling for statistical inference also implies that the standard errors of the coefficients representing trait-environment interaction as estimated by GLM are likely far too small.

Along with the GLM-based tests for association between a single trait and a single environmental variable, we also carried out the tests using the fourth-corner correlation. They resulted in almost identical $p$-values per data set. This finding has led to a companion paper that shows that the squared fourth-corner correlation times the abundance total is a Rao score test statistic on the interaction term in a Poisson log-linear model (*Ter Braak, in press*). This finding was not totally unexpected because Appendix S1 of *Brown et al. (2014)* contains a proof of the equivalence between the fourth-corner correlation and GLM when

trait and environment are factors (although the test statistics are still different, chi-square in fourth-corner and deviance in GLM). The near-equivalence with quantitative trait and environment can be understood mathematically. Under resampling, the term $b_{te} t_j e_i$ is small, so that the expected abundance in the log-linear model

$$\mu_{ij} = R_i^* C_j^* \exp\left(b_{te} t_j e_i\right) \approx R_i^* C_j^* (1 + b_{te} t_j e_i) \tag{6}$$

where the latter can be recognised as the reconstitution formula of a (doubly constrained) correspondence analysis (*Ter Braak, 2014*), which, for a quantitative trait and a quantitative environment variable, is in fact equivalent to the fourth-corner analysis. The test using the fourth-corner correlation is 140 times quicker to compute than the GLM-based test in our R implementation and 1,400 times quicker than the anova.traitglm test. *Peres-Neto, Dray & ter Braak (2016)* already showed that the fourth-corner test is not sensitive to log-linear main effects, thus having power to detect interactions in log-linear models. In the case of single correlations (i.e., pairwise combinations between each trait and each environmental factor), the fourth-corner $p_{max}$ test is convenient and reliable (*Peres-Neto, Dray & ter Braak, 2016*). The GLM-based models were designed for multiple traits and multiple environmental variables. Their advantage and disadvantage relative to rivals such as RLQ, doubly constrained correspondence analysis (*Kleyer et al., 2012*; *Lavorel et al., 1998*) and the combination of site-based and species-based redundancy analyses (*Kleyer et al., 2012*; *Peres-Neto & Kembel, 2015*; *Šmilauer & Lepš, 2014*) need to be investigated.

## ACKNOWLEDGEMENTS

We thank the editor, the reviewers, Emily Charry Tissier and John Birks for their comments that led to a much improved text.

### Funding
The authors received no funding for this work.

### Competing Interests
Cajo J.F. ter Braak is an Academic Editor for PeerJ and the senior author of Canoco, a commercial software package for multivariate (ecological) data analysis and visualization. Stéphane Dray is the author of the R package ade4.

### Author Contributions
- Cajo J.F. ter Braak conceived and designed the experiments, performed the experiments, analyzed the data, contributed reagents/materials/analysis tools, wrote the paper, prepared figures and/or tables, reviewed drafts of the paper.
- Pedro Peres-Neto and Stéphane Dray conceived and designed the experiments, contributed reagents/materials/analysis tools, reviewed drafts of the paper.

## Data Availability

The raw data is available in the R-package ade4 (https://cran.r-project.org/web/packages/ade4/index.html) and in Data S1.

## Supplemental Information

Supplemental information for this article can be found online at http://dx.doi.org/10.7717/peerj.2885#supplemental-information.

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
