# Peer review of "A critical issue in model-based inference for studying trait-based community assembly and a solution"

_PeerJ, doi:10.7717/peerj.2885_

## Round 0.1 · original submission · Minor Revisions

Both the reviewers and I agree that this is manuscript provides a relevant statistical advance for inference of community assembly from traits. The reviewers suggest a number of places in which the manuscript could be improved in clarity, but these constitute only minor revisions of a strong manuscript. I look forward to reading a revision along the lines suggested by the reviewers. In addition, please note that PeerJ does not have a copy-editing service, so please take this opportunity to thoroughly check the manuscript for any typos and grammatical issues.

Best regards

Maria

Reviewer 1 ·

Basic reporting

This manuscript would benefit from major editing on the language front at both broad and detailed scales in order to clearly and impactfully communicate the underlying science.

Results are presented in the introduction (starting on line 102) without methodological context. New tests and results are also introduced in the discussion (line 424, 432, etc.). These are just a couple samples of places the manuscript would benefit from content being reorganized into a more standard, intelligible format.

Sentences (e.g. lines 263 and 264) paragraphs (e.g. lines 223 and 237) and arguably whole sections redundantly repeat content and could be consolidated. In general the paper takes a lot of space to make what is essentially a fairly straightforward set of points, and would be more effective if it were cut substantially in length.

At the fine scale, it is riddled with typos and grammatical errors (e.g. lines 31, 43, 75, 104, 106, 124, 142, 143, 145, 185, 189, 198, 246, etc., etc.). Terms and abbreviations are also repeatedly used without being defined (e.g. line 184, 254).

Experimental design

The experimental design appears sound. While certain aspects of the statistical models employed in the study are beyond my capacity to evaluate, I follow most of the methods and see no major flaws. The discussion touches nicely on caveats that represent potential shortcomings.

Validity of the findings

This paper proposes an advance in the statistical tools used to infer trait-environment relationships from species occurrence data. The pmax test, which had previously been shown to better control error rates (as compared to its one-dimensional alternative) when using the fourth-corner method, is in this paper demonstrated to do the same when using generalized linear regression modeling methods. The validity of this finding is convincing, and represents an important incremental improvement to methods in this field.

It's not clear whether there's sound basis for the recommendation (line 285) that the critical p-value threshold be divided by two -- that 2x scalar value is not demonstrated to be a general feature of the statistical test as opposed to simply an artifact of the particular specification of the simulation in this study. I'd suggest either elaborating on the underlying mathematical reasons that 2 in particular is the appropriate adjustment, or removing the specificity of the recommendation and simply stating that an adjustment may be necessary.

Lastly, if I'm understanding correctly, the "latent variable" issue that's repeatedly discussed (lines 106, 202, 381, abstract) is basically just the "omitted variable bias" that's a fundamental universal limitation of all regression models. If indeed it's just omitted variable bias that's being investigated and discussed here, then it should be identified as such, and should maybe be mentioned only in passing rather than being called out as a key finding since this is just an inherent and widely understood statistical reality that's not unique to this context.

Reviewer 2 ·

Basic reporting

The manuscript is overall clear, and presents the necessary information to demonstrate the importance of the study. Figures are in high quality, and meet the standards of the journal. Minor comments on reporting are provided in the comments to authors. The sections could be better organized to facilitate the understanding. However, the manuscript is relevant and of high quality in its present form.

Experimental design

The research question is clearly defined, and of high relevance for researchers investigating the association of species traits, and environmental conditions on community organization. This also represents original research within the scope of the journal.

Validity of the findings

The analyses and results are sound, and the authors compared the most popular methods currently used for accessing the association of traits/environment/species occurrences. The calculation of p-values using permutation and bootstrapping is also widely accepted. The authors present the conclusions in light of the results found, and the question stated in the introduction.

Additional comments

The manuscript compares the type I error rates of methods testing for the association of species traits and environmental conditions. The authors argue that some models, such as GLMs, that use permutation of sites to calculate p-values will have inflated error, and that randomizing sites and species separately and using the maximum p-value can ameliorate the problem.
The manuscript is an important contribution relevant to the growing body of studies investigating which traits determine the distribution of species.

In spite of the importance of this paper, the overall structure of the manuscript could be improved to facilitate understanding. PeerJ is not interested in the readership or impact, but I believe the authors might be interested in their readers. Therefore, I only have the following minor suggestions:

The illustrative example would be better placed after the simulation, so that the manuscript presents: 1) Introduction section showing the potential problems of current methods, 2) Theory and Methods section about the models used in the manuscript, along with the description of simulations and data used as illustrative example, 3) Results from simulations and from the illustrative example, and 4) Discussion.

In addition, the authors mention functions from the R program in several parts of the manuscript when describing their methods. Although I think this is usually a good practice, the paper would have a broader impact if was not framed specifically to R users. Therefore, the mention of R functions could be limited to description of the methods used (probably only in the Theory and Methods section), and in supporting information.

Below I have other minor suggestions:

l. 5-6: Probably most traits interact with the environment in some way. The key question is to identify the traits that most strongly interact with the environment to determine the distribution of species.

l. 31: Include a reference for Bergmman's rule (ex. Bergmann 1847)

l. 34: be more specific about the “issues” and specify which “complex models” are being referred to.

l. 75: change particular to particularly

l.81-82: Describe how the data was actually simulated in Pollock et al. (2012) (species nor independent units?)

l. 98: detect instead of detects

l. 157-161: The phrase is too long, and difficult to understand. Please, break into two phrases.

l. 176: […] the explanation for the difference […]

l. 180ish: Please, make more clear what the subscripts i and j, and the variables t and e are referring to. Also, consider changing e to another letter to avoid confusion with exponential.

l. 198: Please, just check when you use bootstrap and permutations (and make clear why). In the next section, (l. 252), you mention that permutation is preferred for statistical testing, so it is not clear why you do not use permutation throughout the whole ms.

l. 212: Is the mean of the normal distribution zero? Please, specify.

l.245-251: Please, break this phrase in two.

l. 430-449: Is this paragraph really necessary? Its connection with the rest of the discussion is not very clear.

---

## Round 0.2 · accepted · Accept

Only a few very minor changes were suggested on this round of reviews, which I think can easily be addressed during production.

A great paper. I look forward to seeing it published.

Reviewer 1 ·

Basic reporting

The organization and clarity of the manuscript are much improved from the prior version, and now effectively convey the key research information.

Figure 1 -- this chart has two black solid lines and two black dashed lines. For clarity, it would be good to visually distinguish the horizontal reference lines from the actual data lines. And perhaps to verbally describe the horizontal reference lines in the caption (the solid line clearly seems to be the 0.05 significance threshold, but what's the dashed line? Same question for figure 2).

Experimental design

The research objective is clearly stated, as is its importance to field. The experimental design effectively addresses the issue using a combination of approaches.

Validity of the findings

The findings are sound, and are well communicated in terms of relevance, methods, and context. The authors appropriately discuss remaining areas of uncertainty and needs for further investigation.

Additional comments

This is a nice study that advances methods in an important area of research. I look forward to seeing the paper in print.

Reviewer 2 ·

Basic reporting

No Comments

Experimental design

No Comments

Validity of the findings

No Comments

Additional comments

The authors have made major changes in the manuscript. The paper is much clearer now, and I am certain it will have a broad impact for studies aiming to explain the causes of species abundance based on trait-environment associations.

I have only minor comments that could further improve the manuscript.

Abstract: Remove "for example, due to unobserved traits or environmental variables". The example introduces a break in the middle of a complex phrase what makes it difficult to read.

l. 93: Remove comma after "particularly"

l. 104-114: I really liked the new explanation. It is clear and pleasant to read.

l. 272: "in the anova.traitglm test" would be better than "for..."

l. 322: replace "four-corner" by "fourth-corner"

l. 367: in methods you explain that 30 species and sites were simulated. Include the information about additional simulations in the methods section or point where the results of 100 species are presented.

l. 369: It is not clear why the inflation would be more severe in more complex scenarios. Please, add a phrase to explain this expectation.

l. 411-423 (From "This finding..." to "... equivalent to the fourth-corner analysis"): The comparison with the Rao test does not fit very well here. The discussion would read more smoothly without the phrases from l. 411-423. Perhaps this comparison or some explanation about the method could be included in the methods section.

Thank you for the rewriting. This is an important contribution.